# Association between coordinated counseling from both ASHA and Anganwadi Workers and maternal health outcomes: A cross-sectional study from Madhya Pradesh and Bihar, India

**Lakshmi Gopalakrishnan**[1]*, **Sumeet Patil**[2], **Lia Fernald**[3], **Dilys Walker**[4], **Nadia Diamond-Smith**[5]

1 Institute for Global Health Sciences, University of California, San Francisco, San Francisco, California, United States of America, 2 NEERMAN, Mumbai, India, 3 Department of Community Health Systems, UC Berkeley School of Public Health, Berkeley, California, United States of America, 4 Department of Obstetrics and Gynecology, University of California, San Francisco, San Francisco, California, United States of America, 5 Department of Epidemiology and Biostatistics, Institute for Global Health Sciences, University of California, San Francisco, San Francisco, California, United States of America

* Lakshmi.gopalakrishnan2@ucsf.edu

**Data Availability Statement:** Dataset for this paper publicly available at Harvard Dataverse: https://

## Abstract

Community Health Workers (CHWs) play crucial roles in health promotion and services in rural India. Previous research investigating the effectiveness of coordinated health promotion by different community health workers (CHWs) cadres on health practices is scarce. This study examines the effectiveness of coordinated health promotion by different CHW cadres, specifically Accredited Social Health Activists (ASHA) and Anganwadi Workers (AWW), on maternal health outcomes. Using endline data from a 2019 impact evaluation of 6635 mothers with children <12 months in Madhya Pradesh and Bihar, we compared the association between standalone and coordinated counseling by ASHA and AWW on various maternal health practices. Outcomes included four or more antenatal care visits, birth preparedness, institutional delivery, postnatal care visits, and contraceptive uptake. Fixed effects logistic regression with robust standard errors, corrected for multiple hypothesis tests, was used for analysis. Results showed that 39.6% of women received four or more ANC visits, 31.2% adopted birth preparedness practices, 79.6% had institutional deliveries, 23.3% received postnatal care, and 19.5% adopted a postpartum contraceptive method. Coordinated counseling from both ASHA and AWW was associated with a greater prevalence of four outcomes (birth preparedness, institutional delivery, PNC visit, and contraception) compared to standalone counseling from either ASHA or AWWs. These findings suggest that health promotion by AWW complements that of ASHA, collectively associated with improved health outcomes. This study underscores the effectiveness of coordinated health promotion and highlights the need for multisectoral and coordinated efforts among different CHW cadres at the community level. The results emphasize the importance of integrating various CHW roles to enhance maternal health practices and outcomes in rural India.

**Trial registration number:** https://doi.org/10.1186/ISRCTN83902145.

dataverse.harvard.edu/dataset.xhtml?persistentId=
doi:10.7910/DVN/XKCAIS.

**Funding:** This study is funded by Grant No.
OPP1158231 from the Bill and Melinda Gates
Foundation to the University of California, San
Francisco and University of California, Berkeley.
The funder (Bill and Melinda Gates Foundation)
reviewed and approved the study design, but was
not involved in data collection, data analysis, data
interpretation, or writing of the report. The
corresponding author had full access to all the data
in the study and had final responsibility for the
decision to submit for publication.

**Competing interests:** The authors have declared
that no competing interests exist.

Date of registration: 08/12/2016.

## Introduction

Despite significant progress, maternal mortality and neonatal mortality remain stubbornly
high in India, at 99 per 100,000 and 23.7 per 1000 live births [1–3]. For every one woman who
dies from maternal causes, about 20 women experience morbidities such as postpartum
depression, high blood pressure, and infections [4]. However, a significant proportion of these
maternal, newborn mortality and morbidity can be prevented with improved care in the first
1000 days of life through increased coverage and quality of life-saving interventions such as
antenatal care visits, folic acid supplementation, early initiation of breast feeding, exclusive
breastfeeding, maternal nutrition, social support, among others [5–8]. There is sufficient evi-
dence that community-based delivery platforms connected to local health facilities can expand
coverage of essential interventions as well reduce inequities [5].

In many low-and-middle-income countries (LMICs), including India, community health
workers (CHWs), operate at the frontlines in improving access to and increasing utilization of
essential maternal, newborn and child health interventions through health promotion activities
and counseling, acting as a link between community members and health systems [9–11].
Strengthening CHW programs remains an important strategy in achieving universal coverage of
key community-based evidence—based interventions and health services by the year 2030 [9].
Prior systematic reviews have found that community mobilization, education, and home-based
counseling by CHWs help improve maternal, neonatal and child health outcomes [9, 12, 13].

India has two flagship national programs, the National Health Mission (NHM) and Inte-
grated Child Development Services (ICDS), with combined network of more than 2 million
female CHWs or frontline health workers to deliver maternal health and nutrition promotion
and allied services in rural and remote areas. Accredited Social Health Activists (ASHAs) work
under the NHM of the Ministry of Health and Family Welfare (MoHFW) and Anganwadi
Workers (AWWs) are under the ICDS of the Ministry of Woman and Child Development
(MWCD). The National Health Mission also has another frontline health worker cadre, Auxil-
iary Nurse Midwives (ANMs), based at sub-centre providing services such as health check-up,
referrals, and immunizations with a responsibility of a population of 10000+ from 4–5 villages
to give focused attention to health promotion unlike ASHAs and AWWs who operate at the
village-level [14]. The term "triple A" or "AAA" platform refers to the collaboration or conver-
gence in health services between these three types of CHWs (A: AWWs; A: ASHAs; A: ANMs)
[15]. Besides the health worker system, India has implemented numerous policies and pro-
grams to improve maternal and child health outcomes. Key initiatives include the National
Population Policy (2000), National Health Policy (2002), National Rural Health Mission and
Janani Suraksha Yojana (2005), National Urban Health Mission (2008), Janani Shishu Surak-
sha Karyakram (2011), Rashtriya Kishor Swasthya Karyakram (2014), Pradhan Mantri Surak-
shit Matritva Abhiyan (2016), and Pradhan Mantri Matrutva Vandan Yojana (2017). Notably,
in 2013, India expanded its Reproductive and Child Health (RCH) program into the compre-
hensive RMNCH+A platform, integrating Reproductive, Maternal, Newborn, Child, and Ado-
lescent health services with family planning [16].

Table 1 outlines the overlapping and distinct roles and responsibilities of ASHAs and
AWWs.

Several studies from India have studied the role of CHWs in improving health practices
and outcomes, but not the coordinated counseling or health promotion activities by different

**Table 1. Roles and responsibilities of ASHAs and AWWs to promote the health of mothers and children.**

| | Accredited Social Health Activist | Anganwadi Workers |
|---|---|---|
| Ministry | Ministry of Health and Family Welfare (MoHFW) | Ministry of Woman and Child Development (MWCD) |
| Program | National Health Mission | Integrated Child Development Services |
| Scale | 1.4 million | 980,000+ |
| Education | Completed at least 10 years of schooling | Completed at least 10 years of schooling |
| Population served | Provide services to pregnant and lactating women, adolescent girls and men at the village level, roughly catering to the population of ~1,000 in rural areas. | Provide services to pregnant and lactating women and children 0–6 years, adolescent girls via Anganwadi centers catering population of ~1000. |
| Overlapping responsibilities | • Conduct home visits to promote birth preparedness, institutional delivery, family planning, immunization, regular health check-up, nutrition for mother and child, and breastfeeding.<br>• Record keeping of basic details, services, health and nutrition status, immunizations, etc. on a common ICDS-NHM mother-child protection (MCP) card<br>• Mobilize women and children for monthly Village Health and Nutrition Day and support service delivery (e.g., immunization, health check-up, distribution of iron folic acid, calcium tablets, etc. done by the Auxiliary Nurse and Midwife [ANM]) along with AWWs<br>• Promotion of incentives linked government schemes and benefits<br>• Promotion of water-sanitation-hygiene | |
| Non-overlapping responsibilities | • Accompany pregnant women and children to healthcare facilities for check-up, delivery<br>• promote and accompany women and men for family planning procedures at healthcare facilities<br>• Provide DOTS for patients with TB.<br>• Carry essential provisions (ORS packets, TB medicines, iron and folic tablets, chloroquine<br>• disposable delivery kits, oral contraceptive pills, condoms, sanitary napkins)<br>• Support ANMs in postnatal care check-ups, including homebased newborn care<br>• Universal screening, prevention, and management of non-communicable diseases | • Monthly growth monitoring of children <6 years (height/length, weight, arm circumference)<br>• Referral to ASHA / ANM for health complications, referral of severely malnourished children to nutrition rehabilitation centers.<br>• Community based management of malnourished children<br>• Provide early childhood training to children <3 years and non-formal pre-school education to 3–6 year-old children<br>• Provide monthly take home rations or food supplements to pregnant women, lactating mothers, and children<br>• Provide hot cooked meals at Anganwadi Center to pregnant women and children 6m-6 years<br>• Organize nutrition specific community events such as annaprashan (first solid meal fed to infants), Godbharai (baby shower), and invite ASHA to participate<br>• Carry ORS packets |
| Payment structure | Voluntary worker who receives performance-based incentives based on tasks completed.<br>Four ANC visits (For ensuring antenatal care—INR 300 (USD 3.75) for rural areas; INR 200 for urban areas) + Mobilizing and attending village health and nutrition days (INR 200 or USD 2.50 per session); Institutional delivery (INR 300 or USD 3.75 for rural areas; INR 200 or USD 2.50 for urban areas); Postnatal care (INR 250 or USD 3.00 USD for home visits for postpartum mothers and newborns); Contraceptive method (INR 500 or USD 6.25 for spacing of 3 years after the birth of first child; INR 1000 or USD 12.5 ensuring a couple to opt for permanent limiting method after 2 children; INR 150 or USD 1.8 USD per beneficiary escorting or facilitating beneficiary to the health facility for the PPIUCD insertion. | Part-time contract worker who receives fixed monthly honorarium which varies by states, e.g., USD 55 (INR 4500) in Bihar, and USD 160 (INR 13000 in Madhya Pradesh) |

Note: Compiled from various sources [17–23]. Additional responsibilities, training, and payment amounts of the ASHAs and AWWs may vary by state.

cadres of CHWs together. For example, few studies have identified positive impact of ASHA engagement or contacts in improving maternal health outcomes [18, 24–28]. Studies documenting AWW effectiveness have primarily relied on evaluation of ICDS program. A recent analysis of ICDS led by AWW highlights an improvement in usage of ICDS services between 2006 and 2016, however, the program did not successfully reach households from the poorest backgrounds and women with low schooling levels especially in states with the highest burden of undernutrition [29]. Further, the national survey data also suggests deficiencies in the ICDS

system––only 67% of mothers reported receiving food supplements, growth monitoring, and pre-school education services from AWWs for children below six with wide variation between states [30].

Evidence on coordinated health promotion and integrated service delivery of CHWs between the ICDS and NHM program is scarce. A few studies that have controlled for the presence of other cadre of CHWs did not measure additive effects of receiving counseling from ASHA and AWW together. Rammohan and colleagues examined maternal engagement with any type of CHW cadres (ANM,ASHA,AWW, multipurpose worker) but did not delve into the separate impact of exposure to each of these CHWs [31]. Another study focused on the independent association of antenatal care check-ups provided by ASHAs, AWWs, and any other CHW on birthweight and mortality but did not examine additive association of receiving ANC from all CHWs [28]. Yet another study that examined the association between multiple types of health influences such as ASHA, AWW, ANM, family members, rural practitioners, among others and found that ASHAs were most effective in uptake recommended maternal health behaviors [32]. Only one qualitative study was identified, which explored the role of convergence between ASHA and AWW and underscored the significance of integrated service delivery as a catalyst for increased community participation and accountability [33]. there is a dearth of quantitative research that has systematically examined the effectiveness of coordinated health promotion initiatives led by different types of CHWs in improving maternal and newborn health outcomes.

The lack of evidence regarding coordinated service delivery and health promotion is concerning, particularly in light of the recognition of multisectoral collaboration or convergence of health and nutrition services as an essential strategy for achieving the global goals such as Universal Health Coverage and Sustainable Development Goals [34, 35]. There is growing recognition that the current challenges and inequities experienced by women and children will likely continue to be worsened by the impact of COVID-19, necessitating the imperative for multisectoral collaboration [36]. Moreover, this holds significant importance as the Ministries of Women and Child Development (MWCD) and Health and Family Welfare (MoHFW) have recently acknowledged the importance of collaboration between ASHA and AWW, along with the adoption of a unified behavior change communication strategy. This strategy encompasses joint planning and coordination in areas where the ASHA and AWW responsibilities intersect, with aim of improving maternal and health outcomes [37].

Our study seeks to bridge this evidence gap on the role of coordinated health promotion in context of counseling from both ASHAs and AWWs on maternal health outcomes in India. We hypothesize that coordinated counseling from both ASHA and AWWs will be associated with better health practices, compared to standalone counseling from either ASHAs or AWWs on four or more antenatal check-ups, birth preparedness, seeking postnatal care, adopting postpartum contraceptive method.

## Methods

### Ethics statement

Study protocols were reviewed and approved by institutional review boards at the University of California, Berkeley (Ref. No. 2016-08-9092), and the India-based Suraksha Independent Ethics Committee (Protocol No. 2016-08-9092). The trial is registered at https://doi.org/10.1186/ISRCTN83902145. The procedures for obtaining verbal audio-recorded informed consent from participants before data collection were approved by both the institutional review boards at the University of California, Berkeley and Suraksha Independent Ethics Committee. Verbal audio-recorded informed consent was obtained from the participants prior to data

collection. All methods were carried out in accordance with the Declaration of Helsinki guidelines and regulations.

## Data

This uses secondary data from an endline survey conducted December 2018 through August 2019 across 12 districts of two states of northern India—Madhya Pradesh (MP) and Bihar—to evaluate effectiveness of an mHealth intervention to digitally enable AWWs. More information is available in the published protocol [38] and the final impact evaluation paper [39]. The analytical sample consisted of 6,635 mothers with children <12 months from 852 villages from 12 districts (6 of which had implemented and 6 districts that had not implemented the mHealth intervention) using propensity-score matched design. In each village, the list of beneficiaries available with the AWWs was used as a unit-level sample frame to randomly sample up to eight mothers of children <12 months. All study participants provided verbal audio-recorded informed consent before data collection and after verbally agreeing to proceed with the interview, participants were asked to read or repeat the following sentence: "I have understood the purpose of the interview and my rights as a respondent. I agree to participate in this interview". The consent was recorded on the tablets/devices and the audio recordings of consent were stored securely in compliance with data protection regulations. Before administering the survey, all participants received an information and consent sheet, ensuring they had access to full details about the study. Trained female enumerators surveyed respondents using structured computer-assisted personal interviews. Through the research, the study team only had access to de-identified data which was available to us because our team was responsible for primary data collection. Additional information regarding the ethical, cultural, and scientific considerations specific to inclusivity in global research is included in the S1 Checklist.

## Context of study states

Both Madhya Pradesh (MP) and Bihar face significant health challenges. Under-five mortality rates are high, with 69 per 1000 live births in MP and 60 per 1000 live births in Bihar. Stunting affects 44% of children under five in MP and 49% in Bihar. Anemia prevalence is 55% among pregnant women and children in both states. Educational attainment for women aged 15–49 is low, with only 14% in MP and 12% in Bihar completing 12 or more years of schooling [40]. Antenatal and delivery-related indicators are poorer in Bihar compared to MP. For instance, 36% of mothers in MP had at least four ANC visits, compared to only 14% in Bihar. Institutional deliveries were more common in MP (81%) than in Bihar (64%). However, Bihar outperformed MP in child immunization, with 62% of children aged 12–23 months fully immunized in Bihar compared to 50% in MP [40]. The differences in antenatal care visits and institutional deliveries suggest that MP has somewhat better maternal health service uptake compared to Bihar. This could reflect differences in healthcare access, quality, or health-seeking behaviors between the two states.

## Measures

**Exposures of interest.**   Our main exposures were receipt of *outcome-specific standalone counseling* from ASHAs or AWWs reported by the surveyed mothers during their pregnancy, delivery, and postpartum periods with the recent child. A mother was considered to have received *outcome-specific coordinated counseling* if she reported receiving advice from both ASHA and AWW. Mothers were asked whether and from whom they received any counseling or advice on specific topics such as advice on issues/danger signs and check-up during pregnancy, advice on birth preparedness, advice on institutional delivery, advice on postnatal care

within first 6 weeks of delivery, and advice on family planning. The survey enumerators were trained to specify the life stage related to which the questions were being asked. The survey questions specifically asked the women about advice during each life stage. For instance, *After the delivery of [\*\*INDEXCHILD\*\*], did you receive the following services*? *Advice on family planning(postpartum family planning) or advice on health check-up (postnatal care). From whom*?

**Outcomes of interest.** The outcomes of interest were whether the mother: (i) received four or more ANC check-ups (coded 1 if the woman/mother had at least four ANC visits for index child, 0 otherwise), (ii) adopted birth preparedness practices (was coded 1 if at least 75% of birth preparatory practices were followed by the woman /mother prior to the birth of index child, including identification of health facility for the delivery or in case of emergency during delivery, obtaining clean cloth for drying the baby, obtaining new blade to cut cord, saving money for delivery, arranging transport),, (iii) delivered in a health facility (1 if the woman/ mother reported delivering the index child at a public/private facility, 0 otherwise), (iv) received at least two postnatal check-ups within six weeks of the delivery (1 if woman/mother reported receiving two postnatal check-up at facility within 6 weeks of birth of index child, 0 otherwise), and (v) adopted a contraceptive method post childbirth (1 if the woman/mother reported using any contraceptive method (Female/male sterilization, intrauterine device (IUD), oral contraceptive pills, female / male condoms, injectable contraception, implants, diaphragm, foam/jelly or emergency contraceptive pills, standard days method, lactation amenorrhea, rhythm, withdrawal method) at the time of survey for purposes of preventing pregnancy, 0 otherwise). Exact indicator construction of both exposure and outcome is listed in the S1 Table.

**Covariates.** We control for a several potentially confounding variables as informed by prior literature [18, 26]. Specifically, we control for the following socio-demographic variables in our models: women's age at time of survey (continuous), years of education (continuous), women's work outside the home (yes/no), total number of pregnancies (continuous), and woman's caste. For caste, following the Government of India classifications, we categorized scheduled tribe/scheduled caste as marginalized caste with other backward classes and general caste serving as the reference group. We control for wealth index (in quartiles) from the principal components analysis of household assets [41]. We include fixed effects for the sub-district level to control for any effect of the mHealth intervention. This is important given that the mobile technology intervention was delivered at the district level as well as the sub-district level factors that may potentially affect both service delivery of CHWs and social environment that drives women's health decision making.

## Analytical approach

We describe sample characteristics in terms of frequencies, proportions and/or median and interquartile range (IQR) of the exposure (counseling by ASHAs and AWWs) and outcome indicators and covariates discussed above. For each of the outcomes, including, four or more ANC check-ups, birth preparedness, institutional facility delivery, postnatal care, and contraceptive methods, we fit a fixed-effects logistic regression adjusted for covariates which could potentially bias the estimated association between the outcome and counseling from CHWs. Since the mHealth intervention was assigned at the district level, the fixed effect at sub-district level ensures that the associations are estimated in the original treatment and controlled sub-districts separately and then averaged over. The exclusion of Auxiliary Nurse Midwives (ANMs) from our analysis was a deliberate methodological decision based on the fundamental differences in their role and operational scope compared to ASHAs and AWWs. ANMs

primarily focus on clinical service provision, including health check-ups, referrals, and immunizations, and are responsible for a significantly larger population (over 10,000) spread across 4–5 villages. This broader scope inherently limits their capacity for intensive, village-level health promotion activities [14]. In contrast, ASHAs and AWWs operate exclusively at the village level, allowing them to provide more focused, consistent, and personalized health promotion and counseling services. Their community-embedded nature enables them to engage in frequent, direct interactions with individual families, which is central to the health promotion activities examined in our study.

We quantify the marginal effects (or prevalence differences) and 95% Confidence Intervals (CIs) for each outcome using the *margins* post estimation command in Stata, which uses the delta method to estimate the standard errors of the marginal effect [42]. To test the one-sided hypothesis that effect of coordinated counseling from both AWW and ASHA is larger than that of counseling from either ASHA or AWW, we use one-sided Chi-square tests by using the post estimation *test* in Stata.

All regression models and post-estimation tests used clustered robust standard errors at the village level. Because we assessed association between coordinated counseling for five outcomes separately, we adjusted the p-values for multiple comparisons using Bonferroni correction [43]. We used $p < 0.05$ as statistical significance for all models. All analyses were conducted using Stata Version15 [44].

## Results

Table 2 presents the descriptive characteristics of our study population. On average, women were young with 50% of the sample being below 24 years of age with majority of the population falling between 21 and 26. Despite the young age, the median number of pregnancies was 2, indicating that half of the women in the population have experienced fewer than 2

**Table 2. Socio-demographics of the study population of women with a child 0–12 months (N = 6635).**

|  | n | Value |
|---|---|---|
| Age (median, IQR) | 6,635 | 24 (21–26) |
| Gravida or number of pregnancies (median, IQR) | 6,635 | 2 (1–3) |
| Education in years (median, IQR) | 6,635 | 5 (0–9) |
| **Children age groups** |  |  |
| 0–3 months | 1,533 | 23.0% |
| 3–6 months | 1,957 | 29.5% |
| 6–9 months | 1,719 | 25.9% |
| 9–12 months | 1,426 | 21.5% |
| **Engaged in paid work** | 1,200 | 18.1% |
| **Caste** |  |  |
| Scheduled Caste or Tribe | 2,926 | 44.1% |
| Other Backward Class/General Caste | 3,709 | 55.9% |
| **Wealth quartile** |  |  |
| Quartile 1 | 1,682 | 25.4% |
| Quartile 2 | 1,655 | 24.9% |
| Quartile 3 | 1,652 | 24.9% |
| Quartile 4 | 1,646 | 24.8% |
| **State** |  |  |
| Bihar | 3,417 | 51.5% |
| Madhya Pradesh | 3,218 | 48.5% |

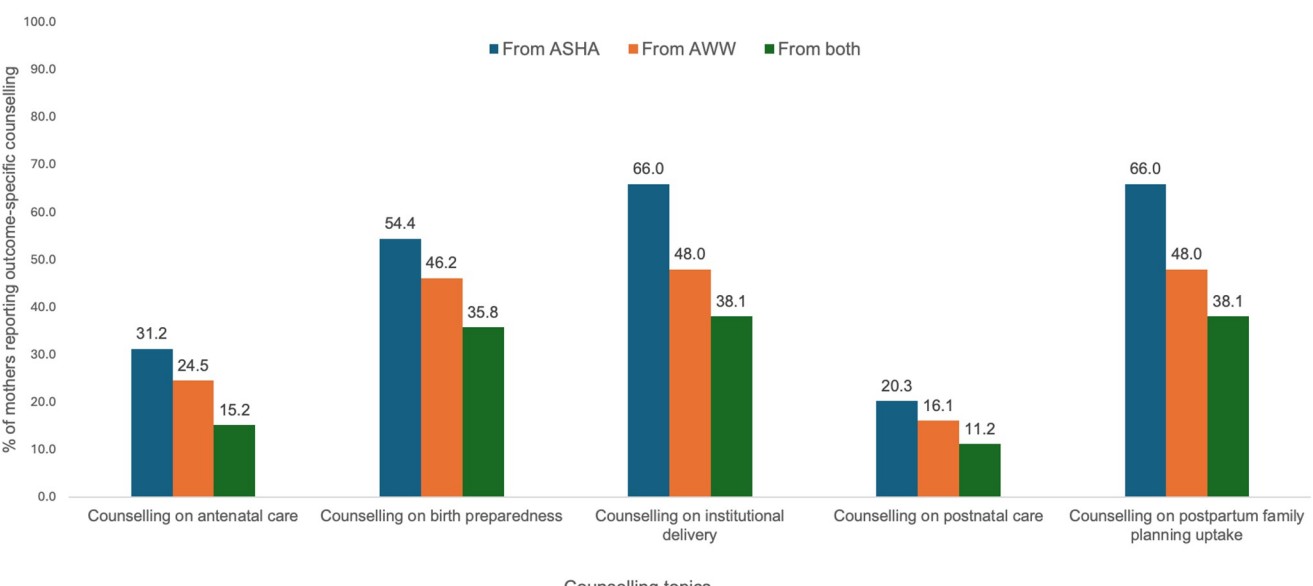

**Fig 1. Coverage of outcome-specific counseling from ASHAs and Anganwadi Workers (AWWs) in Bihar and Madhya Pradesh, N = 6635.**

pregnancies. The level of the educational attainment was low with 50% having fewer than five years of schooling. Almost half (44.1%) of the sample were affiliated with the scheduled caste or tribe. Moreover, a mere 18% of the women reported engaging in paid work outside their households at the time of the survey. The sample was even split between the two states.

Fig 1 presents the proportion of mothers who received outcome-specific standalone counseling from ASHA, AWW or both (coordinated counseling). The most recalled topics of standalone counseling from ASHA and AWW were similar but with higher recall for ASHA––institutional delivery (from ASHA: 66.0%; from AWW: 48.0%), family planning (from ASHA: 66.0%; from AWW: 48.0%) and birth preparedness (from ASHA: 54.4%; from AWW: 46.2%). Coordinated counseling from both ASHAs and AWWs followed a similar pattern––institutional delivery (38.1%), family planning (38.1%), and birth preparedness (35.8%) topics were most recalled by the mothers.

Table 3 presents the results from the main analyses to find association between receipt of outcome-specific counseling from CHWs and the health outcomes. Column (A) presents the mean of the outcomes for *the reference group* of mothers who did not receive any counseling on the given topic from the ASHA or AWW. Columns (B), (C), and (D) present increase in the proportion of mothers reporting an outcome compared to the reference group associated with receipt of the outcome-specific counseling from only the AWW, only the ASHA, or from both.

Contrary to our hypothesis, coordinated counseling on counseling on health issues/danger signs during pregnancy is not statistically significantly associated (-1.3 pp [95%CI: -4.2–1.4 pp]) with mothers receiving four or more ANC check-ups compared to the reference group mean of 36.8%. Standalone counseling from only ASHA or only AWW are also not associated with mothers receiving four or more ANC.

The results are largely as expected in terms of the other four outcomes. The proportion of mothers practicing birth preparedness is substantially higher among those who receive coordinated counseling from both ASHA and AWW (20.5pp [95% CI: 17.3–23.7 pp] compared to reference group mean of 27.2%). The proportion of mothers practicing birth preparedness is

**Table 3. Association between behavior-specific advice from ASHA and Anganwadi Worker (AWW) on maternal, newborn and child health outcomes using logistic regression models† (N = 6635).**

| Health outcomes reported by mothers | Mean in the reference group (mothers reporting no counseling from ASHA or AWW) | Received outcome-specific standalone counseling from ASHA | Received outcome-specific standalone counseling from AWW | Received outcome-specific coordinated counseling from both ASHA and AWW | Comparison of additive effect of coordinated counseling from both with standalone counseling from either ASHA or AWW | |
|---|---|---|---|---|---|---|
| | (A) | (B) | (C) | (D) | (E) | |
| | | Prevalence Difference (B-A) (95% CI) | Prevalence Difference (C-A) (95% CI) | Prevalence Difference (D-A) (95% CI) | H0: $D \leq B$ | H0: $D \leq C$ |
| | | | | | H: $D > B$ | H: $D > C$ |
| | | | | | $Chi^2$ statistics | $Chi^2$ statistics |
| Four ANC visits | 0.368 | 0.009 (-0.024–0.042) | - 0.030* (-0.017–0.058) | -0.013 (-0.042–0.014) | 1.47 | 0.67 |
| Birth preparedness | 0.272 | 0.167*** (0.135–0.200) | 0.097*** (0.051–0.141) | 0.205*** (0.172–0.238) | 5.50** | 26.53*** |
| Institutional delivery | 0.764 | 0.068*** (0.044–0.093) | 0.033** (0.002–0.063) | 0.087*** (0.061–0.112) | 1.5 | 9.83*** |
| Postnatal care | 0.196 | 0.222*** (0.184–0.261) | 0.196*** (0.146–0.245) | 0.292*** (0.248–0.335) | 13.59*** | 29.80*** |
| Contraceptive method | 0.168 | 0.115*** (0.087–0.144) | 0.0770*** (0.038–0.116) | 0.115*** (0.085–0.145) | 0.00 | 5.20* |

*** p<0.01,

** p<0.05,

*p<0.1 adjusted for multiple comparisons using Bonferroni correction; Robust 95%CI in parentheses

†Model adjusted for the following variables: maternal education, maternal age, caste, gravida, household wealth quintile, block-level characteristics.

also higher for standalone messaging from either ASHA (16.7 pp [95% CI: 13.5–20.0 pp]) or AWW (9.7 pp [95% CI: 5.1–14.1 pp]). The coordinated counseling is also more effective than standalone counseling from ASHAs and from AWWs.

In case of institutional delivery, the coordinated counseling is more effective than standalone counseling by AWW, but not more effective than that from ASHA. The association between coordinated counseling and institutional delivery is higher (8.7 pp [95%CI: 6.1–11.2 pp] compared to reference group mean of 76.4%) than standalone counseling from AWW (3.3 pp [95%CI: 0.2–6.3]).

In terms of postnatal care outcome, the association between coordinated counseling and postnatal care check-up within six weeks is also substantially greater (29.2pp [95% CI: 24.8–33.5 pp] compared to reference group mean of 19.6%) than standalone counseling from either ASHA (22.2 pp [95% CI: 18.4–26.1 pp]) or AWW (19.6 pp [95% CI: 14.6–24.5 pp]).

In the case of outcome on postpartum contraception use, the coordinated counseling is more effective than standalone counseling by AWW but not more effective than that from the ASHA. Further, both coordinated counseling and standalone counseling from ASHA were equally associated an increase in the proportion of mothers using a postpartum contraceptive method of 11.5 pp [95%CI: 8.7–14.4] and higher than the association of postpartum contraceptive use with standalone counseling from only AWW (7.7 pp [95%CI: 3.8–11.6 pp).

## Discussion

Overall, our findings indicate that receipt of coordinated counseling from both ASHA and AWW were associated with better uptake of maternal health outcomes. These results were

robust to statistical tests considering Bonferroni corrections for multiple comparison for four out of five outcomes––birth preparedness, institutional delivery, postnatal care check-up and contraceptive uptake.

Our findings suggest that the magnitude of the association between standalone counseling from ASHAs and outcomes were higher than that for the standalone counseling from AWWs. Indeed, all the health outcomes included in this study were core to the NHM and thus led by ASHA, whereas for AWWs, the ICDS nutrition and child growth are the primary outcomes and maternal health outcomes are secondary [45, 46]. ASHAs also get incentives for mobilizing the community for the VHNDs, when pregnant women initiate ANC, for institutional delivery, postpartum home visits, and for promoting family planning methods unlike fixed renumeration for the AWW which can also explain why health promotion by ASHAs has stronger association with health practices [20].

Above findings are consistent with a recent study from rural Bihar that examined the role of multiple health influencers including ASHA, AWW, ANM, family, among others, and found that ASHAs consistently had the strongest association on mothers engaging with recommended health behaviors such as institutional delivery, early antenatal care registration, feeding colostrum and other postpartum behaviors [32] and prior studies documenting the effectiveness of ASHAs [18, 47]. ASHAs remain the primary contact for addressing the healthcare needs of women and children, offering guidance on topics such as birth preparedness, the significance of safe childbirth, and contraception [48]. Further, the performance-linked monetary incentives for ASHAs could be a driving factor in better quality of counseling [20], though we did not have measures for quality of counseling. A review that examined the role of CHWs who provide maternal health services found that that counseling from ASHAs were sought by the beneficiaries as they were more accessible and trusted with community members viewing them as "familiar faces" [49].

Regarding birth preparedness and postnatal care outcomes, our findings underscore the greater effectiveness of coordinated counseling when compared to standalone counseling provided by ASHA or AWW. Specifically ASHAs do not receive incentives for activities associated with birth preparedness practices. Their incentives are tied to promoting institutional delivery, encouraging antenatal care visits during pregnancy, and conducting postnatal care home visits, but not specifically for birth preparedness. Similarly, ASHAs receive incentives for conducting six postpartum home visits, which is distinct from the postnatal checkup at a health facility within the first six weeks. Our study suggests that when ASHA's incentives are not directly linked to a specific health, it appears that coordinated counseling with AWW is more effective. On the other hand, for outcomes such as institutional delivery and contraceptive uptake, where both ASHAs and beneficiaries both receive monetary incentives from the government, standalone counseling provided by ASHA is, statistically, as effective as the coordinated counseling.

In our study, we did not observe a significant association between counseling on check-up during pregnancy and four or more ANC visits. The lack of association between counseling on health issues during pregnancy and mothers receiving four or more ANC check-ups may be explained by the prioritization of counseling topics and the nature of counseling on health issues and danger signs. ASHAs and AWWs might focus more on encouraging institutional deliveries, which they may perceive as more critical for maternal health outcomes, rather than emphasizing the importance of four or more ANC visits. This potential prioritization could be influenced by various factors, including perceived urgency of different health behaviors, time constraints during counseling sessions, or even the structure of incentives for health workers. A national-level analysis of the ASHA program had found that while ASHAs were effective in helping women initiate their maternity care through any ANC, ASHAs were less effective in

ensuring completion of services along the continuum of care such as four or more ANC visits [50]. A modeling study on government payments to ASHAs and actual compensation of ASHAs found that ASHAs were paid for even partial completion of ANC visits, highlighting the importance of correctly aligning incentives in motivating ASHAs to achieve the sequence of desired outcomes, such as four or more ANC visits [51]. Further, prior research has noted many social, cultural economic determinants such as lower maternal education, poor socio-economic status, poorer women's autonomy, and limited spousal support may also disrupt continuity in care, limiting the use of four or more ANC visits [52, 53]. Supply side factors such as low quality or a negative experience with a previous ANC visit may also reduce coverage of later ANC visits [54, 55]. Therefore, even though ASHAs receive incentives when pregnant women initiate ANC, their counseling may not ensure coverage of four or more ANCs and may not be able to overcome the structural barriers. Furthermore, the uptake of 4+ antenatal care (ANC) visits is not solely dependent on women's desire for care or ASHA's counseling effectiveness. It is also contingent on the health system's capacity to provide these services. The consistent presence of Auxiliary Nurse Midwives (ANMs) in villages for at least four Village Health and Nutrition Days during a woman's pregnancy is crucial. This highlights that while demand-side factors are important, supply-side constraints within the health system play a pivotal role in determining ANC utilization.

To our knowledge, this study is among the first to examine the standalone as well as coordinated counseling from multiple cadres of CHWs on maternal health outcomes. Most of the prior studies from India have focused only on either ASHA or AWW counseling and by failing to account for receipt of health and nutrition counseling from the other CHW may have overestimated the effectiveness of a single type of CHW [18, 25, 26, 47]. Some studies have controlled for engagement with other type of CHW [28, 31, 32, 56] but none studied the effect of coordinated health promotion despite their overlapping roles at the community-level.

### Limitations and strengths

Our study has some limitations that most cross-sectional studies face but also some strengths compared to previous evidence on this topic. Our measurements relied on self-reported data by mothers and thus subject to recall bias and social desirability bias. Because the sample for this study is based on propensity score matching and a part of the sample was exposed to an mHealth intervention, the findings are strictly *not* representative of general population of mothers with children <12 months in Bihar and MP, but still based on a large sample of more than 6500 mothers from 841 villages. We draw upon a large and uniquely structured data that encompass a set of outcomes and outcome-specific counseling delivered by ASHAs and AWWs for pregnancy care, childbirth, and postnatal care, thereby enriching the depth and scope of our investigation rather than generic exposure or number of home visits by CHW as have been done in prior studies. The findings of this study are at best associations although we have attempted to control the selection bias to the extent possible with the available data. We acknowledge that our study's scope did not extend to directly examining the role of healthcare workers in the decision-making processes of mothers regarding their care-seeking behaviors. This important aspect of maternal healthcare deserves dedicated attention in future research to provide a more comprehensive understanding of the factors influencing maternal health outcomes.

### Conclusion

Our study adds to the existing literature about how health promotion by CHWs can change maternal health practices. In the context of India, two national-scale programmes (NHM and

ICDS) are coordinating their activities at the community level through their respective CHWs. The coordinated counseling is strongly associated with better health practices except when the practices are depending on external factors and counseling alone is unable to overcome the structural barriers as was the case for four or more ANC visits. Interestingly, our findings suggest that a nuanced relationship between incentive-linked payment models for ASHAs and counseling effectiveness. Health promotion may be more effective when CHWs get outcomes-linked monetary incentives as was the case with ASHA and in such cases, additional effect of counseling by other cadre of CHWs may not be realized. However, when there are no incentives for promoted health practices, coordination between different types of CHWs can be more effective. This also raises important questions about the role of intrinsic motivation and trust in CHW-community relationships. When CHWs counsel on behaviors not linked to incentives, it may reflect a stronger intrinsic belief in the behavior's importance and potentially carry more weight with women, as it is not driven by compensation motives. While financial incentives play a role in shaping ASHA or CHW behavior, factors such as intrinsic motivation, community trust, and health system functionality are equally crucial in determining the program's success in improving community health outcomes [57].

Future research could also explore whether coordinated messaging where both CHWs get outcomes-linked incentives would be more effective, while also considering the potential benefits of non-incentivized counseling in building trust and promoting intrinsic motivation among both CHWs and community members.

## Study implications

The outcomes of our study lend support for promotion of multisectoral collaboration in the form of coordinated counseling by leveraging the complementary roles of ASHAs and AWWs to advance maternal and child health. However, better clarity of roles and responsibilities associated with the CHWs could potentially lead to better service delivery, focusing on a specific set of outcomes [33]. Such clarity in roles may enable India to expedite progress toward achieving universal health coverage in line with the Sustainable Development Goals (SDG) targets.

Future research efforts stand to benefit greatly from the development of standardized definitions and measurement frameworks at a global level for concepts such as coordination, convergence, and collaboration among community-level functionaries and institutions. An example of this is the work being undertaken by Glandon and colleagues in the context of CHWs in India [14]. Additionally, future research should examine the effectiveness of outcomes-linked incentives for AWWs and whether such incentives can complement the health promotion efforts led by ASHAs more effectively. Conversely, it is equally important to investigate whether nutrition-specific outcomes assigned to ASHAs can contribute to improved nutrition outcomes when coupled with counseling provided by AWWs. These studies should also consider the potential long-term impacts of different incentive structures on CHW-community relationships and the overall effectiveness of health promotion efforts.

## Supporting information

**S1 Table. Exposure and outcomes indicator construction.**
(DOCX)

**S1 Checklist. Inclusivity in global research.**
(DOCX)

## Acknowledgments

We extend our sincere thanks to the enumerators at NEERMAN for data collection. Most of all, we thank the survey respondents for their time. We also acknowledge contributions made by colleagues at International Food Policy and Research Institute, NEERMAN, University of California San Francisco, and University of California at Berkeley. We also thank Sneha Nimmagadda at NEERMAN for her efforts in cleaning and preparing the dataset.

## Author Contributions

**Conceptualization:** Lakshmi Gopalakrishnan, Sumeet Patil.

**Formal analysis:** Lakshmi Gopalakrishnan.

**Funding acquisition:** Lia Fernald, Dilys Walker.

**Supervision:** Sumeet Patil, Nadia Diamond-Smith.

**Writing – original draft:** Lakshmi Gopalakrishnan.

**Writing – review & editing:** Lakshmi Gopalakrishnan, Sumeet Patil, Lia Fernald, Dilys Walker, Nadia Diamond-Smith.

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
