## [Decision Letter · Decision Letter 0]

15 Jul 2024

PGPH-D-24-01125

Association between coordinated counseling from both ASHA and Anganwadi Workers and maternal health outcomes in rural India: A cross-sectional study

Dear Dr. Gopalakrishnan,

Thank you for submitting your manuscript to PLOS Global Public Health. After careful consideration, we feel that it has merit but does not fully meet PLOS Global Public Health’s publication criteria as it currently stands. Therefore, we invite you to submit a revised version of the manuscript that addresses the points raised during the review process.

We look forward to receiving your revised manuscript.

Kind regards,

Brian Wahl

Academic Editor

Journal Requirements:

Additional Editor Comments (if provided):

Reviewers' comments:

Reviewer's Responses to Questions

**Comments to the Author**

1. Does this manuscript meet PLOS Global Public Health’s publication criteria? Is the manuscript technically sound, and do the data support the conclusions? The manuscript must describe methodologically and ethically rigorous research with conclusions that are appropriately drawn based on the data presented.

Reviewer #1: Yes

Reviewer #2: Yes

2. Has the statistical analysis been performed appropriately and rigorously?

Reviewer #1: No

Reviewer #2: Yes

3. Have the authors made all data underlying the findings in their manuscript fully available (please refer to the Data Availability Statement at the start of the manuscript PDF file)?

Reviewer #1: Yes

Reviewer #2: Yes

4. Is the manuscript presented in an intelligible fashion and written in standard English?

Reviewer #1: Yes

Reviewer #2: Yes

5. Review Comments to the Author

Reviewer #1: Thank you for submitting the review.

In India, health is a state subject. It would be beneficial to provide a brief overview of the health systems in Bihar and Madhya Pradesh.

Authors completely ignored Health care workers role in Mother care and in decission making process both in Analysis or in limitations.

The use of fixed effects in the analysis was unclear. Consider conducting a stratified analysis comparing intervention and non-intervention areas of mhealth.

Detailed comments are given in the manuscript as comments.

Thank you

Reviewer #2: Thank you for the opportunity to review “Association between coordinated counseling from both ASHA and Anganwadi Workers and maternal health outcomes in rural India: A cross-sectional study”. This study presents quantitative data on the association between counselling by ASHAs alone, AWWs alone, and ASHAs and AWWs both on five maternal and neonatal health-related behaviors (4+ ANC, birth preparedness, institutional delivery, PNC, and contraceptive uptake). The writing is clear and the analysis appears sound. I recommend this paper for publication with minor revisions and consider it a useful contribution to the literature on CHWs, and to our understanding of ASHA and AWW effectiveness and behavior in India.

I present the following minor comments for your consideration.

Please consider introducing and describing the ANM earlier in the manuscript. The ANM should especially be discussed when you mention government efforts to encourage collaboration between ASHAs and AWWs. The official policy that promotes cooperation between the ASHA and AWW also involves the ANM (the “triple A” collaboration), so mentioning the ANM there makes a lot of sense. Consider also explaining earlier in the paper why you did not consider the additive role of the ANM as well.

Could you provide insight into how survey enumerators moved from women’s natural language responses (e.g., “ASHA talked to me after I had my baby”) to specific coded response categories (i.e., was this about postnatal care or postpartum family planning?”).

Please re-read for typos throughout. The “overlapping responsibilities” row is presented twice. “Contrary to our hypothesis, coordinated counseling on counseling on health issues…” Sometimes you write “percent”, sometimes “pp”, sometimes “%”.

Under “payment structure” could you list all the ASHA payments for the health-related behaviors that you focus on in this paper?

In the methods section, could you clarify what behavior women had to report to be considered to have practiced birth preparedness? The other variables are clear to me (4+ ANC, ID, PNC) but birth preparedness seems less black-and-white.

Regarding “Contrary to our hypothesis, coordinated counseling on counseling on health issues/danger signs during pregnancy is not statistically significantly associated (-1.3 pp [95%CI: -4.2 – 1.4 pp]) with mothers receiving four or more ANC check-ups compared to the reference group mean of 36.8 percent. Standalone counseling from only ASHA or only AWW are also not associated with mothers receiving four or more ANC.” � Depending on the exact nature of the “counseling on health issues and danger signs,” ASHAs and AWWs may have presented information to encourage pregnant women to have institutional deliveries rather than to encourage 4+ ANC. This could explain why counseling on danger signs was not associated with ANC uptake.

In regards to “A national-level analysis of the ASHA program had found that while ASHAs were effective in helping women initiate their maternity care through any ANC, ASHAs were less effective in ensuring completion of services along the continuum of care such as four or more ANC visits.” � There is also the issue of whether the 4+ ANC visits were available for women or not. I.e., whether the ANM come to the village for at least four VHNDs during the woman’s pregnancy. Would it make sense to note here that not everything depends on the woman wanting it or the ASHA counseling on it – it also depends on the health system providing it. You rightly mention low quality or negative experiences with ANC, but maybe it is worth considering availability too.

Regarding: “Future research could explore whether coordinated messaging where both CHWs get outcomes-linked incentives would be more effective.” And “Finally, future research should examine the effectiveness of outcomes-linked incentives for

AWWs and whether such incentives can complement the health promotion efforts led by ASHAs more effectively.” � I agree with you. Financial incentives have immediate impact on behavior, particularly for marginalized women who work for very low pay. However I want to suggest up an alternative mentality around counseling and CHW performance, that draws from a framework of intrinsic motivation, worker’s rights, and trust. Might we also consider that when CHWs counsel women to perform behaviors not linked to incentives, the CHW may have a stronger intrinsic belief in this behavior’s importance or the woman may place greater weight on this message (because they know that the CHW isn’t pushing for compensation)? Do we want to break everything down into incentives in order to get CHWs to perform? I am not sure, but I wanted to mention this alternative, and perhaps more critical, perspective for your consideration. You may find this article by Ved and Scott useful (https://www.ghspjournal.org/content/8/3/332.abstract) and may want to bring in a bit more literature on CHW rights and incentives.

Thank you.

6. PLOS authors have the option to publish the peer review history of their article (what does this mean?). If published, this will include your full peer review and any attached files.

**Do you want your identity to be public for this peer review?** For information about this choice, including consent withdrawal, please see our Privacy Policy.

Reviewer #1: **Yes: **Yogish Channa Basappa

Reviewer #2: No

---

## [Decision Letter · Decision Letter 1]

16 Oct 2024

Association between coordinated counseling from both ASHA and Anganwadi Workers and maternal health outcomes: A cross-sectional study from Madhya Pradesh and Bihar, India.

PGPH-D-24-01125R1

Dear Dr. Gopalakrishnan,

We are pleased to inform you that your manuscript 'Association between coordinated counseling from both ASHA and Anganwadi Workers and maternal health outcomes: A cross-sectional study from Madhya Pradesh and Bihar, India.' has been provisionally accepted for publication in PLOS Global Public Health.

Best regards,

Parvati Singh, PhD

Academic Editor

Reviewer Comments (if any, and for reference):

Reviewer's Responses to Questions

**Comments to the Author**

1. If the authors have adequately addressed your comments raised in a previous round of review and you feel that this manuscript is now acceptable for publication, you may indicate that here to bypass the “Comments to the Author” section, enter your conflict of interest statement in the “Confidential to Editor” section, and submit your "Accept" recommendation.

Reviewer #1: All comments have been addressed

2. Does this manuscript meet PLOS Global Public Health’s publication criteria? Is the manuscript technically sound, and do the data support the conclusions? The manuscript must describe methodologically and ethically rigorous research with conclusions that are appropriately drawn based on the data presented.

Reviewer #1: Yes

3. Has the statistical analysis been performed appropriately and rigorously?

Reviewer #1: Yes

4. Have the authors made all data underlying the findings in their manuscript fully available (please refer to the Data Availability Statement at the start of the manuscript PDF file)?

Reviewer #1: Yes

5. Is the manuscript presented in an intelligible fashion and written in standard English?

Reviewer #1: Yes

6. Review Comments to the Author

Reviewer #1: (No Response)

7. PLOS authors have the option to publish the peer review history of their article (what does this mean?). If published, this will include your full peer review and any attached files.

**Do you want your identity to be public for this peer review?** For information about this choice, including consent withdrawal, please see our Privacy Policy.

Reviewer #1: **Yes: **Yogish Channa Basappa
